# Summary of Guidelines for Managing Histoplasmosis among People Living with HIV

**DOI:** 10.3390/jof7020134

**Published:** 2021-02-12

**Authors:** Freddy Perez, Diego H. Caceres, Nathan Ford, Giovanni Ravasi, Beatriz L. Gomez, Alessandro C. Pasqualotto, Paul Hine, Antoine A. Adenis, Mathieu Nacher, Tom Chiller, John Baddley

**Affiliations:** 1HIV, Hepatitis, Tuberculosis and Sexually Transmitted Infections Unit, Communicable Diseases and Environmental Determinants of Health Department, Pan American Health Organization, Washington, DC 20037, USA; ravasigi@paho.org; 2Mycotic Diseases Branch, United States Centers for Disease Control and Prevention, Atlanta, GA 30329, USA; tnc3@cdc.gov; 3Center of Expertise in Mycology Radboudumc/CWZ, 9101 Nijmegen, The Netherlands; 4HIV Department, World Health Organization, 1202 Geneva, Switzerland; fordn@who.int; 5School of Medicine and Health Sciences, Universidad del Rosario, Bogota 111221, Colombia; beatrizlgomez@hotmail.com; 6Universidade Federal de Ciências da Saúde de Porto Alegre and Santa Casa de Misericórdia de Porto Alegre, Porto Alegre 90050-170, Brazil; acpasqualotto@hotmail.com; 7Cochrane Infectious Diseases Group, Department of Clinical Sciences, Liverpool School of Tropical Medicine, Pembroke Place, Liverpool L3 5QA, UK; paul.hine@lstmed.ac.uk; 8Centre d’Investigation Clinique Antilles Guyane, CIC INSERM 1424, Centre Hospitalier de Cayenne, 97300 Cayenne, France; antoine.adenis@gmail.com (A.A.A.); mathieu.nacher66@gmail.com (M.N.); 9Department of Medicine, Division of Infectious Diseases, University of Maryland School of Medicine, Baltimore, MD 21201, USA; jbaddley@ihv.umaryland.edu

**Keywords:** advanced HIV disease, diagnosis, histoplasmosis, *Histoplasma*, guidelines, World Health Organization (WHO), Pan American Health Organization (PAHO)

## Abstract

Histoplasmosis is a frequent fungal opportunistic infection in people living with HIV (PLHIV), associated every year to a total of 5% to 15% of AIDS-related deaths among this population. In 2020, the first global guidelines for diagnosing and managing disseminated histoplasmosis among PLHIV was published. This document recommends (1) detection of circulating *Histoplasma* antigens as the recommended laboratory assay to diagnose histoplasmosis among PLHIV; (2) the use of liposomal amphotericin for induction therapy in severe or moderately severe disease, followed by a maintenance therapy with itraconazole for 12 months; a shorter maintenance therapy could be considered if the patient is clinically stable and if immune status has improved; (3) antiretroviral therapy initiation as soon as possible among patients with histoplasmosis without involvement of central nervous system; and (4) that for the treatment of co-infection with histoplasmosis and tuberculosis (TB), treatment of TB should be initiated according to the World Health Organization treatment guidelines. Appropriate health education of providers, supportive supervision, and policy guidance for the care of PLHIV are required.

## 1. Introduction

Opportunistic infections such as tuberculosis (TB) and severe bacterial, fungal, and parasitic infections are the most frequent causes of death among adults with advanced HIV disease [1,2]. Histoplasmosis caused by the fungus *Histoplasma capsulatum* has been frequently reported in the American continent, where an estimated 15,000 new infections and 9000 deaths occur among people living with HIV (PLHIV); it is also diagnosed in much of the world and presumed to be underdiagnosed in most settings [3,4,5]. Disseminated histoplasmosis (DH), the most frequent clinical presentation of this disease among PLHIV, remains challenging due the non-specificity of symptoms from other infectious diseases, in particular TB, and the limited access to specific diagnostic assays and treatment [6,7,8]. As disseminated histoplasmosis is associated with high mortality rates due to late diagnosis [9,10], rapid detection of histoplasmosis and specific treatment are needed [9,11,12]. The development of a novel histoplasmosis diagnostic assay and its recent inclusion in the WHO list of Essential Diagnostics as well as the inclusion of new effective antifungal agents in the Essential Medicine List, generate new opportunities for improving the management and care of PLHIV having disseminated histoplasmosis [13,14,15]. This review summarizes the recently published World Health Organization (WHO)/Pan American Health Organization (PAHO) guidelines for diagnosing and managing disseminated histoplasmosis among PLHIV [16].

## 2. Materials and Methods

Following WHO processes for guideline development [17], the PAHO/WHO secretariat established a guideline development group (GDG) that included individuals with recognized expertise in the histoplasmosis field. The group was chaired by an expert in the management of histoplasmosis with support from a guideline methodologist. In order to develop systematic reviews and guidelines recommendations, PICO (Patient–Intervention–Comparison–Outcome) questions were proposed (Figure 1).

Systematic reviews were conducted to summarize the evidence relevant to these questions, and the key findings are summarized in the final guidelines [16]. The Grading of Recommendations, Assessment, Development, and Evaluation (GRADE) method was applied to assess the certainty of the evidence as well as to determine the strength of the recommendations [18]. Additional considerations included feasibility, acceptability, supply use and cost implications of implementing the recommendations, and clinical outcomes (such as drug resistance and drug toxicity).

A SWOT (strengths, weaknesses, opportunities, and threats) analysis was conducted. This analysis focused on the following: (1) access to rapid diagnostic assays, (2) access to specific antifungal therapy, and (3) surveillance of HIV-associated histoplasmosis (focusing on morbidity and mortality) [7]. Prior to finalization, the draft guideline was reviewed by an external peer review group.

## 3. Results

The judgments made by the guideline development group for each intervention are summarized in Figure 1.

### 3.1. Diagnosis of Histoplasmosis

**Recommendation 1. “Among people living with HIV, disseminated histoplasmosis should be diagnosed by detecting circulating *Histoplasma* antigens”** (GRADE classified this recommendation as conditional with low-certainty evidence).

A systematic review and meta-analysis compared the accuracy of different diagnostics assays for the detection of histoplasmosis in PLHIV [6]. In this meta-analysis, antigen detection assays showed the highest performance to detect histoplasmosis in PLHIV (pooled sensitivity, 95%, and pooled specificity, 97%) [6,16]. Advantages of this assay include its commercial availability and the ability to be implemented in a laboratory with lower-level biosecurity (levels 1 and 2).

The GDG recommended complementary testing using microscopic analysis of the buffy coat, histopathological analysis, culture, and antibody or DNA detection in patients with clinical forms of histoplasmosis different to the disseminated disease [16,19].

### 3.2. Treatment of Disseminated Histoplasmosis

Three recommendations address the induction and maintenance therapy for histoplasmosis among PLHIV and the clinical stage of the disease (severe or moderately severe and mild to moderately mild).

**Recommendation 2.1A. Induction therapy, for severe or moderately severe disseminated histoplasmosis: “Liposomal amphotericin B, 3.0 mg/kg for two weeks is the preferred treatment for severe or moderately severe disease”** (GRADE classified this recommendation as conditional with very-low-certainty evidence).

Within the systematic review, four studies that provided information about relevant outcomes for induction therapies were identified. This included one randomized control trial that compared liposomal and deoxycholate amphotericin B. Compared to deoxycholate amphotericin B, liposomal amphotericin B may have higher clinical success rates (Relative Risk [RR] 1.46, 95% confidence interval [CI] 1.01 to 2.11; low-certainty evidence). Compared to deoxycholate amphotericin B, liposomal amphotericin B has lower rates of nephrotoxicity (RR 0.25, 95% CI 0.09 to 0.67; 1 study, 77 participants; high-certainty evidence) [20,21,22].

The GDG also recommended that in resource-constrained settings where liposomal amphotericin B is not available, deoxycholate amphotericin B (0.7–1.0 mg/kg) should be used as induction therapy for two weeks in severe or moderately severe DH (conditional recommendation, very low certainty of evidence) [16,20]. In patients with central nervous system involvement, extension of induction therapy or increased amphotericin B dosage could be needed [16].

**Recommendation 2.1B. Treating mild to moderate histoplasmosis: “Itraconazole 200 mg twice daily after a loading dose of 200 mg three times daily for three days”** (GRADE classified this recommendation as conditional with very-low-certainty evidence).

Very-low-certainty evidence was identified to inform comparisons between induction therapy in mild to moderate disease in the systematic review undertaken. Treatment with fluconazole achieved poor results in comparison to other azoles in the single-arm trial [16]. No further trials were done as there was no longer sufficient equipoise to justify this. In conclusion, it was found that triazole drugs present lower efficacy than polyenes among severely ill people. No randomized trial comparing amphotericin B formulations with triazole drugs have been undertaken [20,21,23].

**Recommendation 2.2. Maintenance therapy: “Itraconazole 200 mg twice daily for 12 months is recommended”** (GRADE classified this recommendation as conditional with very-low-certainty evidence). **Less than 12 months of therapy can be considered when the person is clinically stable, receiving antiretroviral therapy, has suppressed viral load, and the immune status has improved** (GRADE classified this recommendation as conditional with very-low-certainty evidence).

In regard to maintenance therapy, the systematic review that was undertaken did not find studies comparing less than 12 months of oral itraconazole or 12 months or greater of oral itraconazole. One prospective single-arm cohort study followed a cohort of participants who discontinued antifungal therapy after at least 12 months, providing the participant had received at least six months of antiretroviral therapy (ART) and had achieved a CD4 count of 150 cells/μL or greater [22]. There were no relapses in 32 participants who discontinued therapy after 12 months. A single, retrospective cohort study provided anecdotal evidence that maintenance therapy could be safely discontinued before 12 months but was at critical risk of bias [20,24]. The recommendation was therefore based on expert opinion.

### 3.3. Timing of Antiretroviral Therapy Initiation

**Recommendation 3. “Antiretroviral therapy should be initiated as soon as possible among people with disseminated histoplasmosis for whom central nervous system involvement is not suspected or proven”** (GRADE classified this recommendation as conditional with very-low-certainty evidence).

Only one randomized clinical trial with 282 participants met the inclusion criteria of a systematic review of studies comparing outcomes of early versus delayed initiation of antiretroviral therapy in patients with disseminated histoplasmosis [25]. Of the total number of participants, 10 had HIV and a presumptive or confirmed diagnosis of disseminated histoplasmosis. By day 30 of follow-up, one of the seven patients in the early antiretroviral therapy group and none of the three in the late group died.

Based on this restrictive evidence, the efficacy and safety outcomes of early versus late initiation of antiretroviral therapy are unknown [16,20,26]. Recommendation is based on the panel clinical expertise and field reports with regard to the balance between the substantial risk of dying from another opportunistic infection when delaying ART and the low incidence of histoplasmosis-associated immune reconstitution inflammatory syndrome (IRIS) versus morbidity and mortality associated with histoplasmosis-associated IRIS among PLHIV receiving ART [26]. However, WHO HIV/AIDS guidelines recommend initiating ART as soon as possible after the diagnosis of HIV (within 7 days) [27]. The panel also recalled that PLHIV presenting for the first time or those returning to care must undergo evaluation for opportunistic infections (TB and cryptococcosis), particularly those affecting the central nervous system, before ART initiation. Immediate ART initiation is not recommended in PLHIV who have cryptococcal meningitis because of the increased mortality presumably related with IRIS [27,28].

### 3.4. Treatment of Co-Infection with Tuberculosis

**Recommendation 4. “People living with HIV with TB and histoplasmosis coinfection should receive TB therapy according to WHO treatment guidelines”** (GRADE classified this recommendation as conditional with very-low-certainty evidence).

Two studies, including one case report describing treatment outcomes among patients coinfected with HIV, tuberculosis, and disseminated histoplasmosis were identified through a systematic review and showed very limited evidence [29,30]. Risks of *Mycobacterium tuberculosis* resistance and drug–drug interactions of a simultaneous rifampicin–itraconazole regimen (leading to sub-therapeutic and ineffective itraconazole blood levels and histoplasmosis treatment failure) were balanced in the context of existing strong WHO’s guidance on managing HIV and TB coinfection [16,20]. The recommendation relies on WHO guidance regarding prompt TB therapy initiation and on the panel’s clinical expertise regarding alternative options to concomitant antifungal therapy with extending the duration of amphotericin B induction therapy, or increasing itraconazole daily dosage as well as monitoring itraconazole blood levels if available, or considering other azole drugs (posaconazole or voriconazole) or replacing rifampicin with rifabutin [16]. Strong consideration was also given to a critical review of potential ART regimen options in the context of concomitant antifungal and anti-TB therapies in order to make necessary adjustments as recommended in WHO’s consolidated guidelines on the use of antiretroviral drugs for treating and preventing HIV infection.

Figure 2 summarizes the guidelines’ recommendations.

## 4. Discussion

This review summarizes the evidence and other considerations, supporting the formulation of WHO/PAHO global recommendations for the diagnosis, treatment, and management of disseminated histoplasmosis and disseminated histoplasmosis–tuberculosis co-infection among PLHIV. These recommendations have been developed through a public health approach taking into consideration contexts with limited resources and a high burden of disease. They are intended for healthcare providers, HIV program managers, policymakers, and other professionals involved in caring for people who either have or may be at risk of developing disseminated histoplasmosis. While the highest burden of histoplasmosis is concentrated in the Americas [4], the recommendations contained within these guidelines are applicable globally and adaptable according to the resources and logistical capabilities.

Early diagnosis and opportune treatment initiation are key to improving mortality from histoplasmosis; for this reason, it is necessary for countries to give priority access to high-quality in vitro diagnostics for the rapid detection of *Histoplasma* antigens [7,11]. Implementing this recommendation infers taking appropriate steps to assure adequate availability of antigen testing in health services managing advanced HIV disease. There is a need to apply price negotiation and pooled procurement mechanisms to improve access, especially where histoplasmosis is highly endemic. Access to optimal antifungal drugs is of great importance. Optimal antifungal medicines for treating people with histoplasmosis (conventional and liposomal amphotericin B and itraconazole) are part of the WHO Model List of Essential Medicines [14]. The cost of treatment is still high, and these drugs are not widely available in many of the countries with a high burden of histoplasmosis. Among the interventions to consider improving the access to antifungal medications for treating histoplasmosis are increased advocacy for drug price reduction and promoting generic production, national registration of all antifungal drugs, applying joint regional procurement mechanisms, and developing proper drug forecasting and monitoring systems. Furthermore, appropriate health education for providers, supportive supervision, and policy guidance at the national level on managing histoplasmosis among people living with HIV are required.

The systematic reviews highlighted that overall, the certainty of the evidence for most outcomes was low or very low in this field. Non-randomized study designs predominate in this area of research, many of which were compromised by uncontrolled confounding. This led to the development of a number of priority research recommendations [7]. The process has offered an opportunity to set a research agenda and thereby accelerate the development of new and improved diagnostics and optimize program implementation.

## Figures and Tables

**Figure 1 jof-07-00134-f001:**
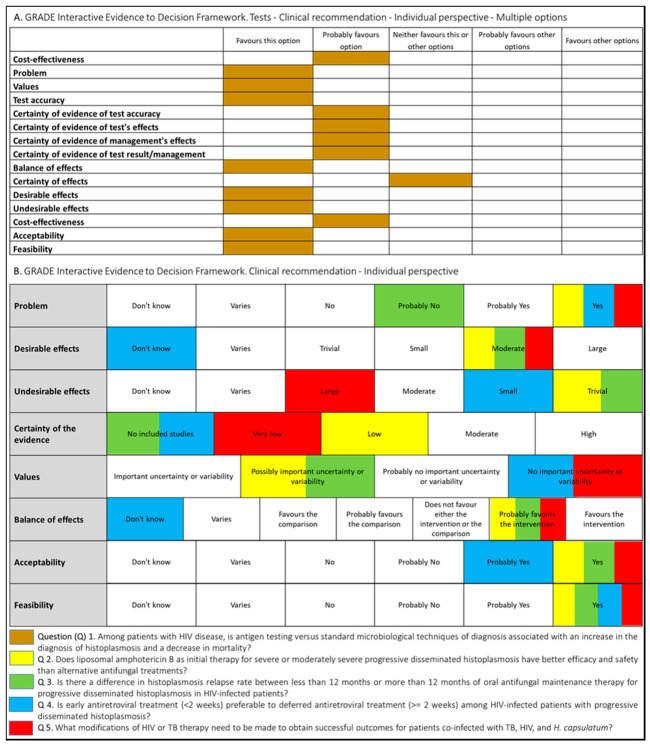
Summary of judgment of questions proposed for the development of the guidelines for diagnosing and managing disseminated histoplasmosis among people living with HIV.

**Figure 2 jof-07-00134-f002:**
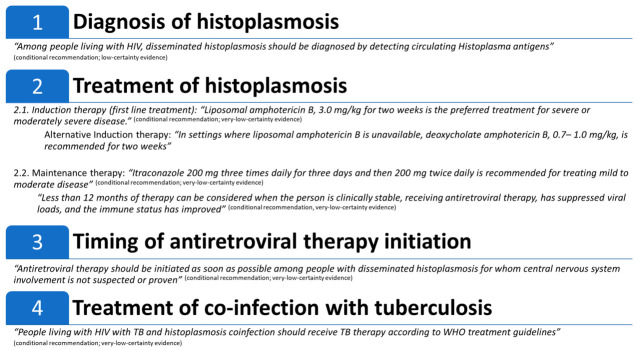
Summary of: Pan American Health Organization (PAHO)/World Health Organization (WHO) guidelines for diagnosing and managing disseminated histoplasmosis among people living with HIV.

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
