# Peer review of "Summary of Guidelines for Managing Histoplasmosis among People Living with HIV"

_jof, 2021, doi:10.3390/jof7020134_

Round 1

Reviewer 1 Report

The paper "Summary of guidelines for managing histoplasmosis among people with HIV" by Perez et al well summarizes how is the best management to diagnose and treat histoplasmosis among PLHIV, answering to the four most important questions concerning the diagnostic approach, the therapeutic protocol and the concomitant antiretroviral and TB treatments.

It is essential and clear, so it can be an useful document for health education of providers and policy guidance on managing histoplasmosis.

Author Response

  1. Thank you for your comments.

Reviewer 2 Report

Perez, et al. present a summary of the guidelines for diagnosis and treating histoplasmosis in people living with HIV. The manuscript is generally very well written and useful, but one has to wonder what is the value of a summary document/review document when the primary document has been printed. The answer to that, would be to annotate and comment on the primary recommendations, especially paying attention to controversies and gaps, and giving expert advice since the authors are indeed experts in the field. Some items that should be addressed:

  1. Preferred term throughout is people living with HIV.
  2. Mention the role/value of the Buffy coat test.
  3. Expand on the different formulations of itraconazole and their nuances.
  4. Expand on timing of antiretroviral therapy initiation and make concrete recommendations.
  5. Greatly expand the section on coi-infection with Tb and give very specific recommendations for the concomitant use of azoles and rifamycins.

Author Response

Perez, et al. present a summary of the guidelines for diagnosis and treating histoplasmosis in people living with HIV. The manuscript is generally very well written and useful, but one has to wonder what is the value of a summary document/review document when the primary document has been printed. The answer to that, would be to annotate and comment on the primary recommendations, especially paying attention to controversies and gaps, and giving expert advice since the authors are indeed experts in the field. Some items that should be addressed:

R: We appreciate this concern to avoid unnecessary redundancy. The motivation for developing this manuscript was to succinctly summarize the recommendations and underlying evidence and key considerations in the medical literature. This will support dissemination of the key information to a different audience and actors, compared to the full guideline product. It is anticipated that this will support implementation of the recommendations, documentation of impact, and research conduct to answer unaddressed questions. 

Thank you for your valuable comments and suggestions.

1.Preferred term throughout is people living with HIV.

R. We reviewed the entire document and added the complete term where it was missing. In general, it is preferred to avoid the word patient to describe an individual with a manageable chronic condition. For this, we have also reviewed the document throughout to ensure that the term ‘patient’ is also used appropriately.

2.Mention the role/value of the Buffy coat test.

R. We modified the final statement of the first recommendation where we state this test. Additionally, we added a reference to support this that describes the role of the Buffy coat to diagnose histoplasmosis “Leitão et al. J Fungi (Basel) . 2019 Jun 9;5(2):47. doi: 10.3390/jof5020047.”

3.Expand on the different formulations of itraconazole and their nuances.

R. the WHO/PAHO guidelines did not specify different itraconazole formulations, reason why we prefer to keep the recommendation as it is presented in the guidelines.

4.Expand on timing of antiretroviral therapy initiation and make concrete recommendations.

5.Greatly expand the section on coi-infection with Tb and give very specific recommendations for the concomitant use of azoles and rifamycins.

R: We appreciate these two suggestions (#4 and 5), but it is not possible to address these points. The recommendations were formulated through a rigorous process based on up-to-date systematic evidence reviews and other information interpreted by an expert panel. We do not feel it would not be appropriate for this group of authors to extend/modify what was developed through the rigorous guideline process.

Round 2

Reviewer 2 Report

The authors have carried out minor corrections in response to my review, but have failed to embrace suggestions that would greatly enhance the value of the manuscript.

Author Response

We  modified statements in the ARV and TB co-infection sections as recommended.
